**Data Availability Statement:** All relevant data are within the supporting information

**Funding:** The author(s) received no specific funding for this work.

# Health anxiety, coping mechanisms and COVID 19: An Indian community sample at week 1 of lockdown

**Evelyn Barron Millar[1‡], Divya Singhal [2‡], Padmanabhan Vijayaraghavan[2‡], Shekhar Seshadri[3], Eleanor Smith[4,5], Pauline Dixon[6], Steve Humble[6], Jacqui Rodgers[1], Aditya Narain Sharma [4,5]***

**1** Population Health Sciences Institute, Faculty of Medical Sciences, Newcastle University, Newcastle Upon Tyne, United Kingdom, **2** Goa Institute of Management, Sanquelim, Goa, India, **3** National Institute of Mental Health and Neurosciences, Bengaluru, India, **4** Translational and Clinical Research Institute, Faculty of Medical Sciences, Newcastle University, Newcastle Upon Tyne, United Kingdom, **5** Cumbria Northumberland Tyne and Wear NHS Foundation Trust, Newcastle Upon Tyne, United Kingdom, **6** School of Education, Communication and Language Sciences, Faculty of Humanities and Social Sciences, Newcastle University, Newcastle Upon Tyne, United Kingdom

‡ These authors share first authorship on this work.
* aditya.sharma@ncl.ac.uk

## Abstract

It is critical to gain an understanding of the impact of the COVID 19 pandemic and the associated lockdown restrictions on the psychological, social and behavioural functioning of the general public, in order to inform public health promotion and future health service resource allocation. This cross-sectional study, completed during week 1 of lockdown in India, reports on data from 234 participants using an online survey. Data regarding health anxiety, coping mechanisms and locus of control was collected. Structural equation modelling was used to assess the relationship between locus of control, coping mechanisms, health anxiety and age. Age related differences in both locus of control and coping strategies were found. Younger people experienced more health-related anxiety and were more likely to engage with social media as a coping mechanism. Mindfulness-based strategies may reduce health anxiety by increasing tolerance of uncertainty experienced during the COVID 19 pandemic.

## Introduction

The initial focus during the COVID 19 pandemic has been on its physical health sequalae and the public health interventions required to minimise transmission. Existing literature indicates that pandemics have the potential to have an impact on psychological, social and occupational function [1, 2]. There is emerging literature to suggest that public health interventions such as *lockdowns* may add further to this dysfunction [3]. It is, therefore, critical to understand more about these impacts and the coping mechanisms being used by different groups in society. This knowledge could be used to inform public health promotion and future health service resource allocation.

**Competing interests:** The authors have declared that no competing interests exist.

Previous research following the SARS and Ebola outbreaks in 2003 and 2014 respectively found widespread fear induced emotional reactions that impeded infection control [4, 5]. In a recent systematic review [6] high rates of mental health symptoms were reported in studies assessing the general population in China, Spain, Italy, Iran, USA, Turkey, Nepal, and Denmark during the first 6 months of the COVID-19 pandemic. These mental health symptoms included anxiety (6.33% to 50.9%), depression (14.6% to 48.3%), post- traumatic stress disorder (7% to 53.8%), non-specific psychological distress (34.43% to 38%), and stress (8.1% to 81.9%). Recent studies have also reported that increased exposure to social media and/or news information concerning the pandemic was positively associated with increased symptoms of anxiety [7–9]. These data are in keeping with earlier reports of increased stress and vulnerability at the population level in response to previous outbreaks, particularly in those who are less resilient [10–15].

Locus of control, defined as an individual's belief or perception about the source of consequences in life [16], may influence levels of response during times of crisis. Individuals with an internal locus of control may perceive that consequences result from personal actions, whilst individuals with an external locus of control may perceive consequences as caused by external events largely out of their control and influence [17]. Those with an internal locus of control tend to exhibit higher resilience during times of uncertainty [18, 19]. Literature suggests that the internal locus of control may increase until middle age, decreasing thereafter [20, 21].

Elevated health related anxiety is understandable during a pandemic. As stated in the literature around health anxiety [22–24] the stress due to COVID 19 is situated in the feelings of contracting or having the disease. The triggering of high levels of anxiety and the ensuing coping mechanisms during an event such as the current global pandemic, is aligned with the transactional model of stress and coping as set out by Lazarus and Folkman [25]. This model indicates that behavioural and cognitive coping responses are used by individuals in order to control internal and external stressors. For some individuals these anxiety symptoms may be inherently frightening and/ or misperceived as signs of physical illness, thus increasing overall anxiety levels. When faced with threat social resources can play an important role in an individual's coping mechanism and ultimately their ability to function [25, 26]. Of course, the pandemic is an inherently uncertain situation and research indicates that difficulties coping with uncertainty [known as intolerance of uncertainty - 27] are an important transdiagnostic mechanism in a range of anxiety disorders, including health anxiety [28]. Writing about the 2009 H1N1 pandemic Taha and colleagues report that greater intolerance of uncertainty was related to lower appraisals of self- and other control, which in turn was associated with low rates of problem-focused coping and greater reports of pandemic related anxiety [10]. Furthermore, they report that people with high levels of intolerance of uncertainty were more likely to perceive the pandemic as threatening and were more likely to use emotion-focused coping strategies, and both of which predicted elevated levels of anxiety. Similarly, in a recent study by Rettie and Daniels, conducted in the context of the COVID 19 pandemic, it was demonstrated using mediation modelling that maladaptive coping responses partially mediated the relationship between intolerance of uncertainty and both anxiety and depression scores [29].

In an effort to reduce the anxiety and gain certainty about health status, individuals will often engage in safety-seeking behaviours, including searching for health related information on the internet [30]. Whilst of course the use of safety behaviours in the presence of actual threat, such as a pandemic, is essential to maintain survival, excessive and inflexible use of such behaviours has been observed to maintain anxiety disorder symptoms [31]. Safety behaviours associated with health anxiety include seeking reassurance from external sources, including doctors, social media and internet searches [30]. These behaviours may be employed by an individual to reduce the perception of threat which in turn may create a short-term reduction

in health anxiety [32].However such safety behaviours are negatively reinforcing and maintain anxiety symptoms in the long-term [33] and maintain and exacerbate beliefs that health-related uncertainty cannot be tolerated [34], creating a self-perpetuating cycle of health anxiety symptoms [35].

Traditionally, responses to public health crises focus on containment, vaccine development, distribution of drugs and economic revival, with psychological and social needs receiving less priority, both in terms of a delay in ascertaining the burden and allocating necessary resources [36]. This is even more relevant in a developing country such as India as seen during the Bhopal gas tragedy. In December 1984, following a leakage of a highly hazardous chemical, the local population reported symptoms of suffocation, intense irritation, and vomiting. Murthy highlights the significance of this disaster thus [37]:

"The Bhopal disaster is of importance in the relevant literature for a number of reasons. First, it is one of the largest man-made disasters in a developing country. Second, the disaster effects were a combination of both the chemical substances inhaled and the psychological effects. Third, no formal mental health infrastructure was available to provide post-disaster mental health care, and this led to the development of the innovative approaches to care. Fourth, this disaster has been the subject of intense study, both cross-sectionally and longitudinally, from physical and mental health viewpoints."

The tragedy highlights the importance of both identifying and then treating mental health sequalae of emergencies in developing countries. This is also, highlighted in the WHO Mental Health Gap Action Programme (mhGAP) which aims to scale up services in low- and middle-income countries as there is a lack of access to appropriate treatments [38]. Knowledge of the psychological, behavioural and social responses is vital not only to understand this aspect of the burden posed by COVID 19 but also to inform effective public health interventions and future mental health service provision.

This study aims to explore the psychological impact of COVID 19 on health anxiety and the coping strategies used by the adult population in India during the first week of lockdown as a result of COVID 19 pandemic. Our overall conceptual model is taken from the core ideas of the transactional model of stress and coping as set out by Lazarus and Folkman [25]. Our conceptual model sets out the empirical relationships among the antecedent, mediating and outcome variables that make up the health anxiety process (Table 1).

Hypothesis: Age impacts the mediating relationship of COVID 19 health anxiety between locus of control and coping mechanisms.

**Table 1. Model of health anxiety process.**

| Causal antecedent → | Mediating process→ | Outcome |
|---|---|---|
| Locus of control | Health anxiety | Coping |
| **Internal** | Worry | Behavioural |
| Personal responsibility | Risk | |
| Social responsibility | Vulnerability | |
| Values and commitments | Preoccupation | Mindful |
| | Distress | |
| **External** | Somatic anxiety | |
| Beliefs | | Social |
| Control | | |

Source: Adapted from Lazarus and Folkman (1987).

## Materials and methods

This cross-sectional study is based on data collected by researchers from Goa Institute of Management, India. The Centre for Excellence in Research at the Goa Institute of Management, India was consulted regarding the research protocol. As the survey was anonymous, institutional ethics approval was not required. The Qualtrics link was shared through social media (WhatsApp and Facebook), participants were advised of the purpose and nature of the study. No incentives were offered for participation. Particular reference was made to the importance of studying the impact of COVID 19 on their levels of health anxiety, well-being and coping mechanisms. Participants were informed that the survey would take approximately 10 minutes to complete. Participation in the study was voluntary and anonymous with no personal identifiable data captured. Participants were not permitted to skip any of the items; however, they had a choice to abandon their response at any point.

Data were collected in the English language during week 1 (24th March– 30th March 2020) of lockdown in India. The survey consisted of 23 items capturing four constructs: *Health Anxiety (*6 items), *Coping Mechanisms*, *Internal Locus of Control, and External Locus of Control* described in Table 2. All responses were made on a 5-point Likert type scale, ranging from 1 (strongly disagree) to 5 (strongly agree).

**Table 2. Measures.**

| **Health anxiety scale (adapted from the Short Health Anxiety Inventory [39])** |
|---|
| 1. I have started to feel worrisome about my own health. |
| 2. I have started to feel I am at risk for developing illness |
| 3. I wonder what body sensations/change mean to me |
| 4. I am becoming aware of bodily sensations/changes |
| 5. I have started to feel worrisome about my own loved one's health |
| 6. I frequently read/hear about illness and think I have it |
| **Coping mechanisms (adapted from ways of coping check list (WCCL) [40])** |
| 1. I am thinking of learning something new |
| 2. I feel happy for having more time to be with my family |
| 3. I spend time reading |
| 4. I am taking a good rest |
| 5. I'm practicing meditation to help me cope |
| 6. I have started exercises and yoga at home |
| 7. I am having an ability to resist thoughts of illness |
| 8. My social media usage has gone up |
| 9. I'm using technology to connect with my loved ones |
| 10. I read and enjoy humorous messages and share with others |
| **Locus of control (adapted from the Rotter internal-external locus of control scale [16]** |
| **Internal** |
| 1. I feel it is my responsibility to follow government precautions |
| 2. I feel responsible for my health and my loved ones |
| 3. I feel responsible for neighbours and society |
| 4. I feel I am integral part of the preventive mechanism |
| **External** |
| 5. Cure from Covid-19 is a matter of good luck |
| 6. Getting this disease is a matter of bad luck |
| 7. The prevention of Covid-19 is against anyone's control |

In order to ensure all measures were appropriate to the sociocultural setting in which the study was conducted, the following steps were followed: Health Anxiety scale, 4 behavioural experts from different institutes including Goa Institute of Management were asked to suggest appropriate items from the Short Health Anxiety Inventory (Abramowitz et al. 2007a) to capture health anxiety. Based on the responses, 5 items were re-phrased from the inventory and 1 item (I have started to feel worrisome about my own loved one's health) was added as a new item. For the Coping Mechanisms and Locus of Control items the same experts were asked to select the items in a similar way. Repetitive items were excluded and some of the items were re-phrased for easy reading.

A hybrid model with both structural and measurement components was analysed to assess the relationship between locus of control, coping mechanisms and health anxiety. This structural analysis was subsequently explored with three path models to illustrate how factors change with age [41–43].

Structural Equation Modelling (SEM) was carried out in Stata to establish the construct validity of the survey (Fig 1). A range of fit and comparison-based indices, including chi-square, were used to determine which model provided the best fit for the data [44–46] The fit indices included Root Mean Square Error of Approximation (RMSEA), Standardised Root Mean Square Residual (S-RMR), Coefficient of Determination (CD), Tucker-Lewis Index (TLI) and Comparative Fit Index (CFI). Hu and Bentler [47] suggest various cut offs for these fit indices. To minimize Type I and Type II errors a combination with S-RMR or the RMSEA were used. In general, good models should have an S-RMR <0.08 or the RMSEA <0.06 with the fit index values > 0.9.

## Results

There were 234 participants (Male:Female = 148 (63%): 86 (37%)) with 101; (43%) aged 18–31 years, 75 (32%) aged 32–45 years and 58 (25%) aged 46 years and above. The majority of the respondents (141, 60%) were from non-metro cities (a metro being defined as a city having a population of 1 million or more). One hundred and ten participants (47%) were employed; 30 (13%) had their own business, 55 (23%) were students and the remaining 39 (17%) were

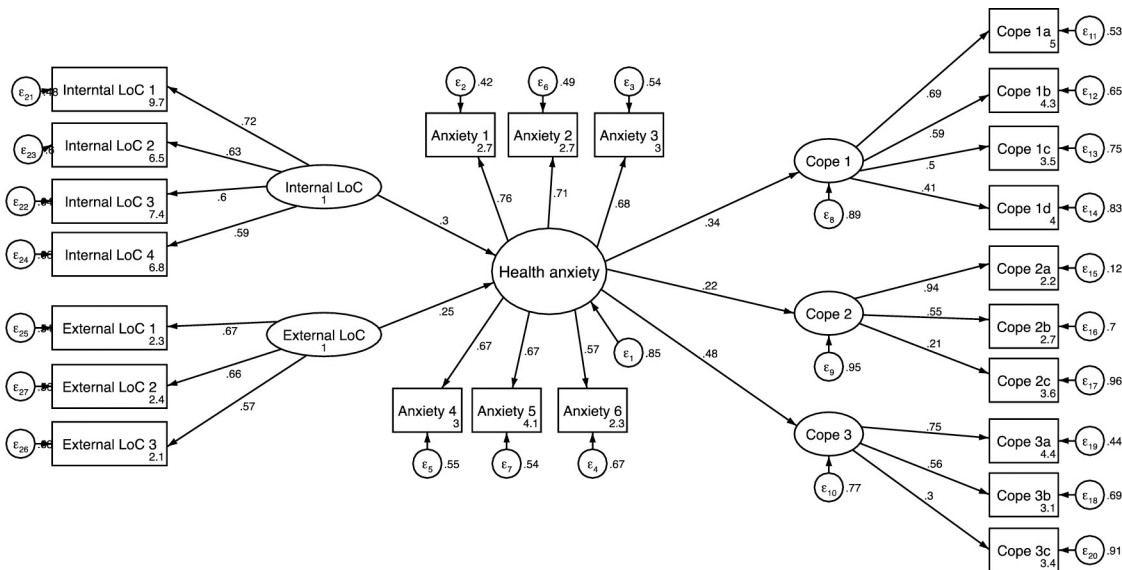

**Fig 1. Hybrid model with structural and measurement components.**

retired or homemakers. Two hundred and nine participants (89%) were living with their families during the lockdown period.

The internal consistency of the measures were:

- Health Anxiety Scale α = 0.838

- Coping mechanisms α = 0.651.

- Internal locus of control α = 0.717.

- External locus of control α = 0.673.

Exploratory factor analyses were undertaken on the 10 coping items in order to test for the smallest number of interpretable latent factors. An initial estimation yielded 3 factors with eigenvalues exceeding unity, accounting for 52.6% of the total variance. The internal consistency and the items within each factor are listed below:

Cope 1: Behavioural coping (α = 0.623)

- I am thinking of learning something new

- I feel happy for having more time to be with my family

- I spend time reading

- I am taking a good rest

Cope 2: Mindful Coping (α = 0.522)

- I'm practicing mediation to help me cope

- I have started exercises and yoga at home

- I am having an ability to resist thoughts of illness

Cope 3: Coping through social media (α = 0.511)

- My social medium usage has gone up

- I'm using technology to connect with my loved ones

- I read and enjoy humorous messages and share with others

As the Cronbach alpha internal consistency measures are poor to questionable further analysis was performed to determine the reliability [48]. Fig 1 details the hybrid model which included both structural and measurement components to assess the relationship between locus of control, coping mechanisms and health anxiety.

Information RMSEA, S_RMR, CD, TLI and CFI of the individual measures are provided in Table 3 (Model I), demonstrating that the predicted model is an acceptable fit [49, 50].

It was considered whether any revisions to this model should be made given the acceptable fit indices. Modification indices recommended the inclusion of some disturbance covariance terms between item variables. This resulted in a better data to model fit (Model II) and confirmed our confidence in the structure.

**Table 3. Fit Indices of the whole model.**

| | $\chi 2$ | df | RMSEA | S-RMR | CD | TLI | CFI |
|---|---|---|---|---|---|---|---|
| | | | | Fit Index | | | |
| Model I | 429.934 | 225 | 0.063 | 0.08 | 0.920 | 0.812 | 0.833 |
| Model II | 194.749 | 194 | 0.004 | 0.055 | 0.926 | 0.999 | 0.999 |

**Table 4. Regression standardised structural coefficients.**

| Exogenous variable | Endogenous variable | | | |
|---|---|---|---|---|
| | Health Anxiety | Behavioural Coping | Mindful Coping | Coping through Social Media |
| Internal LoC | | | | |
| Direct effect | 0.304*** | | | |
| Indirect effect | | 0.102* | 0.067* | 0.146** |
| External LoC | | | | |
| Direct effect | 0.247*** | | | |
| Indirect effect | | 0.083* | 0.054* | 0.119* |
| Health Anxiety | | | | |
| Direct effect | | 0.336*** | 0.220** | 0.480*** |

p***<0.001

p**<0.01

p*<0.05. Measurement path coefficients all sig p<0.05.

Reliability is also illustrated by the items which loaded well on to each of the factors, health anxiety (0.76 to 0.56), Cope 1: Behavioural Coping (0.69 to 0.41), Cope 2: Mindful coping (0.94 to 0.21), Cope 3: coping through social media (0.75 to 0.30), internal locus of control (0.72 to 0.59) and external locus of control (0.67 to 0.57). Loadings greater than or equal to 0.3 are said to be salient, relating meaningfully to a primary or secondary factor (Brown, 2006). The multiple regressions for the standardized structural model estimates of the direct effects are summarized in the table below (Table 4). There are significant indirect effects mediated by health anxiety from internal and external locus of control to all of the coping strategies as illustrated in the table above. This indicates that a person's internal and external locus of control have an effect on how they cope.

In order to examine the associations between the 6 latent variables, bivariate correlations were conducted (Table 5). Internal and external locus of control were negatively correlated. All of the coping strategies were positively correlated with each other. Two coping strategies, Cope 1: Behavioural Coping and Cope 3: Coping through Social Media were positively correlated with internal locus of control. Cope 2: Mindful Coping was positively correlated with external locus of control.

**Table 5. Bivariate correlations between the latent variables.**

| | Internal Loc | External Loc | Anxiety | Cope 1: Behavioural Coping | Cope 2: Mindful Coping | Cope 3: Coping through Social Media |
|---|---|---|---|---|---|---|
| Internal LoC | 1 | | | | | |
| External Loc | -0.148* | 1 | | | | |
| Anxiety | 0.192** | 0.135* | 1 | | | |
| Cope 1: Behavioural Coping | 0.319** | -0.131* | 0.214** | 1 | | |
| Cope 2: Mindful Coping | 0.042 | 0.194** | 0.157* | 0.226** | 1 | |
| Cope 3: Coping through Social Media | 0.230** | -0.015 | 0.284** | 0.248** | 0.173** | 1 |
| SD | 1.161 | 1.213 | 1.087 | 1.219 | 1.174 | 1.200 |

All factor scores have a mean of zero.

p**<0.01

p*<0.05.

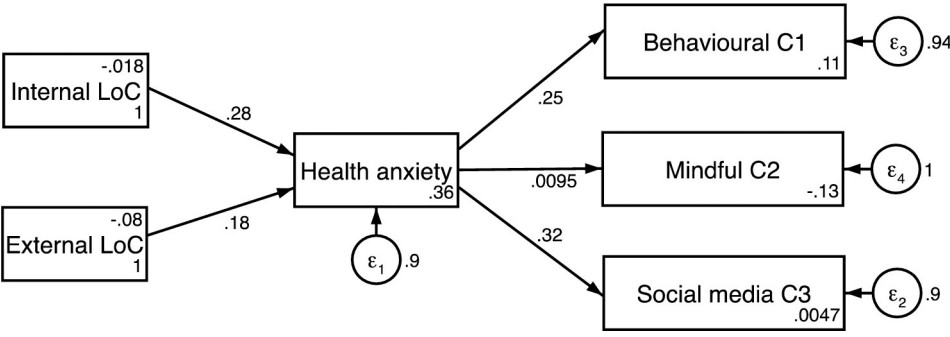

**Fig 2. Structural path model for 18–31 years.**

Regarding sample size a desirable goal for an excellent model is to have a ratio of 20:1 for the number of respondents to the number of parameters in the model. A good model should have a ratio of 10:1 and if the ratio is less than 5:1 the model is said to be poor (Humble, 2020; Kline, 1998). With 6 parameters in the model (internal LOC, external LOC, Anxiety, Cope 1, Cope 2, Cope 3) then the ratios would be excellent for 120 respondents and good for 60 respondents. The categories in this model have the following number of respondents illustrating a good sample size for each of the models:

- 18 to 31 years -101 respondents

- 32 to 45 years– 75 respondents

- over 46 years– 58 respondents

Using these three age categories, 3 standardized coefficient structural path models using the latent factor scores from the proceeding analysis were constructed (Figs 2–4). The model used measured scores for each construct formed by the refined Bartlett method [46, 51, 52]. A test for measurement invariance was undertaken on age subgroups comparing the unconstrained model with all parameters free across groups, to one with coefficients invariant. The chi-square and fit indices difference ($\Delta\chi^2(14) = 10.942$, p = 0.691; $\Delta$RMSEA = 0.036; $\Delta$CFI = 0.027) demonstrates the models measurement invariance [53, 54]. The equality of the model was also assessed through metric invariance tests. These tests of the structural model gave non-significant results across all structural path parameters indicating that the constructs within this model hold across groups [55, 56].

Results indicated participants aged 32 and 45 years old had a balanced locus of control with equal weighting given to both internal and external locus of control ($\beta = 0.2$, CI = 0.03 to 0.42

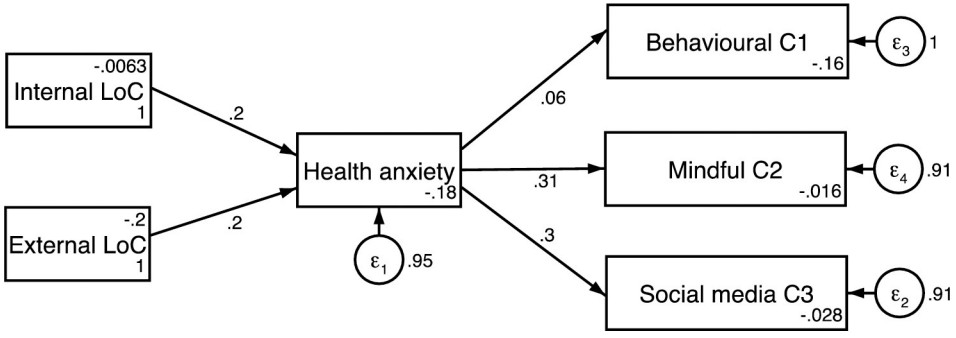

**Fig 3. Structural path model for 32–45 years.**

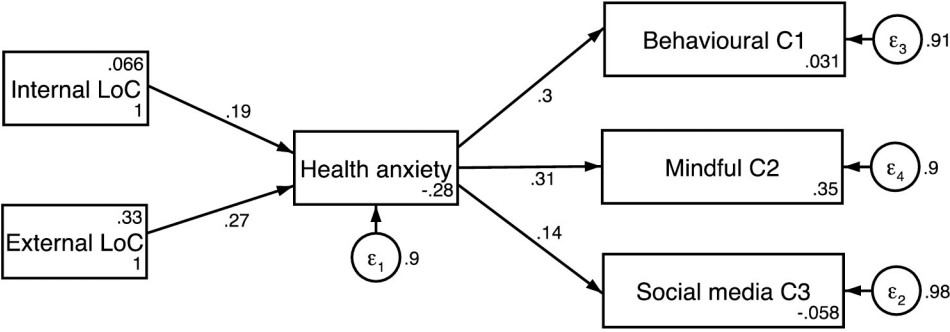

**Fig 4. Structured path model for 46 years and older.**

and β = 0.2, CI = -.02 to 0.43, p<0.05). This was not replicated in the other age groups. The model for the 18–31-year-old group showed a model weighted towards an internal locus of control (β = 0.28, CI = 0.10 to 0.45, p<0.01) and the model for the 46 years and over group was weighted towards an external locus of control (β = 0.27, CI = 0.03 to 0.50, p<0.05).

For the coping strategies, the model for the 46 years and older group was weighted nearly equally between Cope1: Behavioural coping (β = 0.30, CI = 0.07 to 0.53, p<0.01) and Cope 2: Mindful Coping (β = 0.31, CI = 0.07 to 0.53, p<0.01) but not Cope 3: Coping through Social media' (β = 0.14, CI = -0.11 to 0.40, p>0.05). The model for the 18–31-year-old group was weighted towards Cope 3: Coping via Social media (β = 0.32, CI = 0.14 to 0.49 p<0.001) and to a slightly lesser degree Cope1: Behavioural Coping (β = 0.25, CI = 0.07 to 0.43, p<0.01) but not Cope 3: Mindful Coping (β = 0.0095, CI = -0.19 to 0.20, p>0.05). The model for the 32–45-year-old group were nearly balanced for Cope 2: Mindful Coping (β = 0.31, CI = 0.10 to 0.51, p<0.01) and Cope 3: Coping through Social Media (β = 0.30, CI = 0.10 to 0.51, p<0.01) but not Cope 1: Behavioural Coping (β = 0.06, CI = -0.17 to 0.29, p>0.05).

The linear regression analysis shows that there were 3 significant variables related to health anxiety (Table 6). Results indicated that younger age was predictive of greater health anxiety associated with COVID 19 (β = -0.233, p<0.001). The coping mechanism Cope 3: Coping through Social media also significantly predicted health anxiety in response to COVID 19 (β =

**Table 6. Linear regression.**

| Coefficients[a] | | | | | |
|---|---|---|---|---|---|
| Model | Unstandardized Coefficients | | Standardized Coefficients | t | Sig. |
| | B | Std. Error | Beta | | |
| (Constant) | .354 | .224 | | 1.580 | .115 |
| Age of the participants | -.312 | .083 | -.233 | -3.752 | .000 |
| Gender (female = 1) | .019 | .135 | .009 | .141 | .888 |
| Participant lives with their family | -.154 | .211 | -.044 | -.730 | .466 |
| Participant from a metro or non-metro city | .105 | .133 | .048 | .788 | .431 |
| Internal Locus of Control | .120 | .061 | .129 | 1.970 | .050 |
| External Locus of Control | .187 | .056 | .210 | 3.321 | .001 |
| Cope 1: Behavioural Coping | .101 | .060 | .114 | 1.691 | .092 |
| Cope 2: Mindful Coping | .083 | .060 | .090 | 1.379 | .169 |
| Cope 3: Coping through Social Media | .173 | .057 | .192 | 3.020 | .003 |

a. Dependent Variable: Exploratory health anxiety latent variable. $P[F_{(9, 224)} = 6.328] < 0.001$], $R^2 = 0.203$.

0.192, p<0.001). Finally, external locus of control also significantly predicted health anxiety (β = 0.210, p<0.01).

## Discussion

This study explored the impact of the COVID 19 pandemic on health anxiety, reported locus of control and the coping strategies used by adult population in India during the first week of lockdown. The novel findings included age related differences in both locus of control and coping strategies. Data regarding locus of control (internal vs external) as predictor of anxiety in the structured path model were in keeping with previous literature [20, 21] which suggests that internal locus of control may increase until middle age and decrease thereafter. In addition, health-related anxiety was highest in the 18–31 years group which has been previously reported [6, 57, 58]. It has been suggested when facing a problem, younger adults are more likely to react to it with active emotions such as anxiety or fear that motivate them to solve the problem than do older adults [59].

Moving to patterns of coping strategies, the use of social media predominated in the 18–31 years group. The 32–45 years group used mindful activities and social media and the 46 years and older age group used behavioural and mindful coping strategies. The increased use of social media could be viewed as an avoidant coping mechanism [40]. On the other hand, it could be that this coping strategy allows these individuals to seek out social and emotional support to alleviate anxiety [60, 61] Although helpful in the short term, any gains would be limited as the feeling of relaxation would end once social media use is finished. The reliance on turning to social media as a reference point and a coping mechanism, places those using this strategy at greatest risk of anxiety by perpetuating a sense of uncertainty, confusion and helplessness. Social media by its very nature lacks clarity, is contradictory and perpetuates 'fake news' [62, 63]. The use of behavioural and mindful coping strategies may help alleviate anxiety, through a process akin to acceptance and active engagement, supporting previous research on the putative benefits of mindfulness [64] and Acceptance and Commitment Therapy (ACT) [65] for health anxiety. Indeed Kraemer and colleagues report higher levels of mindfulness may result in internal experiences to be perceived as less threatening, thereby increasing one's ability to tolerate uncertainty and decreasing the need to engage in safety behaviours, such as information seeking, seen in the youngest group, that may maintain health anxiety [66].

### Limitations

The findings set out in this paper are preliminary, and further work is required to fully understand the differential impact of these different coping strategies on health and wellbeing and the role of uncertainty. In addition, the small sample size limits the generalisability of the findings. The participant recruitment using social media and the requirement to complete survey in English language are further limitations the study. *A number of scales were adapted for this study. Although the internal consistency for these measures is encouraging, cross cultural adaptation and validation is yet to be undertaken.*

### Conclusions

The data used for the analysis give a unique insight into levels of health anxiety and patterns of coping strategies in an adult Indian community sample to the COVID 19 pandemic. The findings also provided important insights into age related differences in response to the pandemic and some preliminary insight into the effectiveness of differential coping mechanisms to mitigate against the anxiety inevitably conferred by this global crisis. It has been shown that

younger people were experiencing more health-related anxiety in response to COVID 19 and were more likely to engage in coping through social media than their older counterparts. Tentative support for previous work indicates that mindfulness-based strategies may reduce risk for anxiety potentially by increasing tolerance of the inevitable uncertainty currently experienced. The uniqueness of this global pandemic and the gathering of data in this India context around coping will provide an important point of reference that enhances understanding and knowledge relating to mental health strategies. With a limited number of studies around COVID 19 emanating from India with a focus on mental health makes this an internationally important study.

As Murthy says, 'though disasters are a challenge in every country, for the affected populations as well as the mental health professionals, they represent special challenges and opportunities in developing countries' [67]. The Bhopal disaster was the first disaster in India to be studied systematically for the mental health effects with three decades of mental health initiatives. The trauma of the survivors of the Bhopal gas disaster has continued. In the absence of systemic intervention, the impact on the mental health of the affected residents has been reported to have worsened involving both neurotoxicity and a range of mental disorders [68]. An important lesson learnt is that there is a need for continuous dialog with the survivors, to both understand their perceptions and needs, as well as for wide use of "self-care" measures for mental health. The findings of this study echo the significance of this lesson.

## Supporting information

**S1 Data.**
(XLSX)

## Acknowledgments

The authors would like to acknowledge the North East England South Asia Mental health Alliance as a platform for this research consortia. They would also like to thank Mrs Kathryn Parker for administrative input.

## Author Contributions

**Conceptualization:** Divya Singhal, Padmanabhan Vijayaraghavan, Steve Humble, Aditya Narain Sharma.

**Formal analysis:** Pauline Dixon, Steve Humble.

**Investigation:** Aditya Narain Sharma.

**Supervision:** Aditya Narain Sharma.

**Writing – original draft:** Evelyn Barron Millar, Divya Singhal, Padmanabhan Vijayaraghavan.

**Writing – review & editing:** Divya Singhal, Padmanabhan Vijayaraghavan, Shekhar Seshadri, Eleanor Smith, Pauline Dixon, Steve Humble, Jacqui Rodgers, Aditya Narain Sharma.

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
