## [Decision Letter · Decision Letter 0]

17 Dec 2020

PONE-D-20-23653

Health anxiety, coping mechanisms and COVID 19: an Indian community sample at week 1 of lockdown

PLOS ONE

Dear Dr. Sharma,

Thank you for submitting your manuscript to PLOS ONE. After careful consideration, we feel that it has merit but does not fully meet PLOS ONE’s publication criteria as it currently stands. Therefore, we invite you to submit a revised version of the manuscript that addresses the points raised during the review process.

We look forward to receiving your revised manuscript.

Kind regards,

Johnson Chun-Sing Cheung, D.S.W.

Academic Editor

PLOS ONE

"For this anonymised survey institutional approvals were sought as appropriate."

b. Once you have amended this statement in the Methods section of the manuscript, please add the same text to the “Ethics Statement” field of the submission form (via “Edit Submission”).

3. Please provide additional details regarding participant consent.

In the ethics statement in the Methods and online submission information, please ensure that you have specified (a) whether consent was informed and (b) what type you obtained (for instance, written or verbal, and if verbal, how it was documented and witnessed). If your study included minors, state whether you obtained consent from parents or guardians. If the need for consent was waived by the ethics committee, please include this information.

5. We note you have included a table to which you do not refer in the text of your manuscript. Please ensure that you refer to Table 5 in your text; if accepted, production will need this reference to link the reader to the Table.

**Comments to the Author**

Reviewer #1: Dear Editor(s),

Thank you for the opportunity to review this manuscript. This research seems timely, but having read the manuscript, the study seems like it would benefit from a much more careful consideration of relevant bodies of literature, as well as the articulation of a coherent theoretical/conceptual framework and hypotheses/research questions. Several things could also be done to strengthen the “Materials and Methods” section.

Below I draw your attention to some of my major concerns regarding this manuscript, which are followed by some comments by section and page. Should you have any questions or concerns about my review, do not hesitate to get in touch.

Major Concerns

1-One of my chief concerns regarding the manuscript pertain to what I perceive to be a lack of theoretical depth. The study is loosely drawing on a very limited number of sources to make an argument, but there’s no overall, coherent theoretical model or framework guiding the investigation. And, of course, there are no hypotheses or research questions stated.

2-On a related note, the authors are testing a model via structural equation modeling procedures, but have not presented and justified an initial conceptual model.

3-There are several important pieces of information missing from the section on “Materials and Methods”:

>It is not clear how participants were recruited. The authors mentioned recruitment via social media, but how was the study presented to prospective participants? Were there incentives offered? Moreover, how do the demographics of this convenience sample compare to official demographics for the population they come from? Why is the nature of the sample not discussed as a limitation later in the paper?

>Additionally, for two of the three measures adapted from other studies, there are no citations.

>Further, for the measure that the authors constructed themselves – the one on “Coping” – there is no analysis of validity reported. There is a Cronbach alpha score reported, which seems quite low (the author indicate in “Limitations” that the internal consistency results are encouraging, but the findings do not quite support this conclusion). There’s no explanation from the authors for the low numbers. Moreover, where did these dimensions of coping that the authors have identified come from? Why is there no reference to the coping-related literature in psychology, public health, communication/media, etc. (with the exception of one to Folkman et al.’s work in the Discussion section)? The authors seem to talk at some length about social media use in the Conclusion, but have made no reference to the literature on social media use.

>Plus, there is no justification for the inclusion of all the variables mentioned in Table 5, reporting findings from a linear regression predicting health anxiety.

>Additionally, the authors make two references to the Bhopal disaster, which occurred back in 1984, but never provide any additional information about it; so international audiences may not be able to appreciate these references and the significance of that tragic event.

Additional Comments by Section and Page (page numbers refer to the pages of the PDF)

Section: Introduction

p. 9

“In the current pandemic, data from China indicate severe mental health symptoms in the affected regions (Dong and Bouey, 2020), fuelled by vast amount of information available via social and other media sources (Mertens et al., 2020)”: symptoms such as? What do these studies find? These seem like very general statements.

“Heightened exposure to pandemic related information may influence risk perception and play a role in shaping coping mechanisms (Gilman, 2010)”: Play a role how? And what is it about “heightened exposure” that matters here?

p. 10

“Unfortunately these behaviours can in turn negatively reinforce health anxiety and beliefs that health-related uncertainty cannot be tolerated (Olatunji et al., 2011), creating a self-perpetuating cycle of health anxiety symptoms (Abramowitz et al., 2007b)”: This is an over-generalization. This could happen, but obviously not for everyone. In many cases health-related information seeking can help reduce uncertainty, too, no? I wouldn’t base my argument on just one study.

p. 10

“This is even more relevant in a developing country such as India as seen during the Bhopal gas tragedy in 1984 (Murthy, 2014)”: Why more relevant in a developing country?

Section: Materials and Methods

p. 11

“For this anonymised survey institutional approvals were sought as appropriate”: Sought, but were they actually obtained? If the survey was anonymous, this study would likely get “exempted” status, so I am just suggesting clarification here.

p. 11

Was the measure for “Coping Mechanisms” which was created by the authors examined for its validity? If so, how? What dimensions of coping are captured? There’s a significant literature on coping that has not been consulted/used here.

Also, what is the source for the “Locus of Control” measure? Please provide citations.

Section: Findings

p. 16

How did these age groups get defined (e.g., 18-31, 32-45, 46+)? Based on what?

Reviewer #2: 

First impressions of the study:

The research appears to be taking initial steps to identify mental health factors related to the COVID-19 pandemic and initial stages of lockdown conditions in India. There is limited data on this phenomenon globally and would add to the growing database – especially as preliminary findings from South Asia. The study has potential to add to international research, and the contribution of knowledge to facilitate comparisons between South Asia and rest of the world with reference to coping mechanisms and health anxiety. The manuscript appears to be relevant and interesting.

Title, Abstract, References:

Full title is informative, relevant, to the point and conveys key features of the article. The title is likely to spark interest in the scientific mental health community – given the present context of living with COVID-19 conditions and the need to inform provisions for services in mental health in South Asian and Globally.

The objectives of the study are clearly stated within the abstract, with methods outlined, and results and conclusions section which is clearly aligned with aim. The methods section within abstract can be improved by indicating the measures used within the study.

References cited and listed correctly. Recommended further inclusion of recent international research articles on mental health related to coping and health anxiety within discussion during COVID -19 as points of comparison or discussion would improve impact of article. Recent literature has been published related to COVID-19 and health anxiety experienced globally, for e.g. Germany (Jungmann & Witthoft, 2020), UK (Rettie & Daniels, 2020), Turkey (Ozdin & Ozdin, 2020), US (Tull, et al., 2020), Canada (Taylor, Landry, Paluszek, Rachor, & Asmundson, 2020).

Introduction:

The research topic has been introduced by placing the study well within the research context with adequate literature. The importance of the current research has been established and the purpose of the study has been clearly stated. However, the first study aim appears to be general and needs to be specified and revised to state “the study aims to explore the psychological impact of COVID-19 on health anxiety”.

Logical progression in flow of thought observed with enough information to understand the research. Specialized terminology such as locus of control has been explained, and statement of population has been provided. Structure and quality of writing is satisfactory.

Methodology:

The process of study subject selection is clearly provided, however, it was observed that ethical clearance to conduct study was not obtained by the authors. Manuscript states that institutional clearance was obtained – although not specifically stated from where. Please provide this information for clarity of ethics approval for study conducted in the field of psychology/psychiatry among human participants.

The procedure does not indicate the language in which the study was conducted – this needs to be indicated for reproducibility of the study and also to correctly gauge study population and thereby, any implications. Further, the measures listed in study gives rise to the requirement of following clarifications,

a. Is the provided reference correct for the ‘Health anxiety scale’? – The citation given refers to the Psychometric properties of the Short Health Anxiety Inventory, and not the six-item scale used in this study.

b. Was the health inventory adapted/validated to the Indian context? If so, the correct reference should be provided.

c. Was there a method used to generate the questionnaire items for the scales?

d. How was the sample size estimated for this study?

Minor errors- Typos observed in item 5 ‘mediation’ and item 8 ‘social medium’ under the coping mechanisms scale.

Results:

In general, data has been presented well, with the text provided adding to the tabulated/illustrated results. If there is a recognized cut off mark for health anxiety it is suggested to indicate the percentage and levels of health anxiety observed in the study sample within descriptive data.

In process of answering the study questions, the authors have also taken efforts towards the establishment of construct validity of the author generated question items/factors. However, correct interpretation in reporting encouraged. For instance, internal consistency reliability for the author generated question items in coping mechanisms and locus of control are less than 7, which is questionable. Further, the three factors, within the coping scale (cope 1,2 and 3) also report alpha values less than 7 and 6, which are usually interpreted as ‘poor to questionable’ (George & Mallery, 2003).

Line 103, the authors provide values to what figured constitute a good fit. However, model fit has been declared a ‘good’ fit, whilst the findings are not reporting the same. The model fit are adequate or borderline RMSEA > 0.06, S.RMR = 0.008 (not less), CFI <0.9. Consider revising to state, ‘acceptable fit’ should the literature support the findings (Byrne, 2016) (Kline, 2016).

Within the linear regression data, it was observed gender was not included as an independent variable. Consider including as the relationship with gender and health anxiety would also add to the available database on the subject.

Discussion:

Meaningful results have been discussed well and placed within context without over interpretation. Conclusions answer the aims of the study (aim one in introduction section needs to be specified). Limitations of study have been clearly indicated. Following minor modifications are suggested,

a. Line 211 - … “the use of other ‘adaptive’ behavioral and mindful techniques..”

b. 235-240 – Revise and summarize the final paragraph to increase impact.

c. Line 200 – addition of other references that are also confirming that young adults such as (Gerolimatos & Edelstein, 2012) report increased levels of health anxiety would further support the findings.

Overall comments:

Study design poses limitations however, acceptable. Rigorous statistical analysis conducted to establish construct validity of measure, and results for the study. The study confirmed coping patterns and health anxiety relationship with age, and contributes novel information regarding mental health impact during COVID-19 from South Asia. Article is consistent with its self.

References

Byrne, B. M. (2016). Structural Equation Modelling with AMOS: Basic Concepts, Applications, and Programming. New York: Routledge.

George, D., & Mallery, P. (2003). SPSS for Windows step by step: A simple guide and reference (4th ed., Vol. 11.0). Boston: Allyn & Bacon.

Gerolimatos, L. A., & Edelstein, B. A. (2012). Anxiety-related constructs mediate the relationbetween age and health anxiety. Ageing and Mental Health , 975-82.

Jungmann, S. M., & Witthoft, M. (2020). Health anxiety, cyberchondria, and coping in the current COVID-19 pandemic: Which factors are related to coronavirus anxiety? Journal of Anxiety Disorders , 73.

Kline, R. B. (2016). Methodology in the social sciences.Principles and practice of structural equation modeling (4th ed.). Guildford: Guildford Press.

Ozdin, S., & Ozdin, S. B. (2020). Levels and predictors of anxiety, depression and health anxiety during COVID-19 pandemic in Turkish society: The importance of gender. International Journal of Social Psychiatry .

Rettie, H., & Daniels, J. (2020). Coping and tolerance of uncertainty: Predictors and mediators of mental health during the COVID-19 pandemic. American Psychologist .

Taylor, S., Landry, C. A., Paluszek, M. M., Rachor, G. S., & Asmundson, G. J. (2020). Worry, avoidance, and coping during the COVID-19 pandemic: A comprehensive network analysis. Journal of Anxiety Disorders , 76.

Tull, M. T., Barbano, A. C., Scamaldo, K. M., Richmond, J. R., Edmonds, K. A., Rose, J. P., et al. (2020). The prospective influence of COVID-19 affective risk assessments and intolerance of uncertainty on later dimensions of health anxiety. Journal of Anxeity Disorders , 75.

---

## [Author Response · Author response to Decision Letter 0]

29 Jan 2021

Editor Comment No1. 

Authors’ response No 1:

We would like to thank the editor for the reminder and links. Changes made as advised.

Editor comment No 2. 

Thank you for including your ethics statement:

b. Once you have amended this statement in the Methods section of the manuscript, please add the same text to the “Ethics Statement” field of the submission form (via “Edit Submission”).

Authors’ response No 2: 

Thank you for drawing this to our attention. On pages 7-8 lines 133-142 now read as follows: ‘The Centre for Excellence in Research at the Goa Institute of Management, India was consulted regarding the research protocol. As the survey was anonymous, institutional ethics approval was not required. The Qualtrics link was shared through social media (WhatsApp and Facebook), participants were advised of the purpose and nature of the study. No incentives were offered for participation. Particular reference was made to the importance of studying the impact of COVID -19 on their levels of health anxiety, well-being and coping mechanisms. Participants were informed that the survey would take approximately 10 minutes to complete. Participation in the study was voluntary and anonymous with no personal identifiable data captured. Participants were not permitted to skip any of the items; however, they had a choice to abandon their response at any point.’

Editor comment No 3. 

Please provide additional details regarding participant consent.

In the ethics statement in the Methods and online submission information, please ensure that you have specified (a) whether consent was informed and (b) what type you obtained (for instance, written or verbal, and if verbal, how it was documented and witnessed). If your study included minors, state whether you obtained consent from parents or guardians. If the need for consent was waived by the ethics committee, please include this information.

Authors’ response No3: 

Participation was voluntary as outlined above. In advance of competition of survey, all participants were provided with information about the purpose and nature of study on social media, including required time commitment. If participant was happy to proceed, they then clicked on included link to access survey. 

All participants were over the age of 18 years and no minors were included in study. Lines 171-172 of manuscript read as ‘There were 234 participants (Male:Female=148 (63%): 86 (37%)) with 101; (43%) aged 18-31 years, 75 (32%) aged 32-45 years and 58 (25%) aged 46 years and above.’

Editor comment No 4. 

We note that you have indicated that data from this study are available upon request. PLOS only allows data to be available upon request if there are legal or ethical restrictions on sharing data publicly. For more information on unacceptable data access restrictions, please see http://journals.plos.org/plosone/s/data-availability#loc-unacceptable-data-access-restrictions. 

Authors’ response No 4: 

Data file in excel format has now been uploaded to Plos One with these amendments. 

Editor’s comment No 5:

We note you have included a table to which you do not refer in the text of your manuscript. Please ensure that you refer to Table 5 in your text; if accepted, production will need this reference to link the reader to the Table.

Authors’ response No 5: 

Thanks for bringing this to our attention. The table is now table 6 (page 15) and reference to the same has been made in the manuscript as advised (line 263). 

Comments to the Author

Reviewer #1: Dear Editor(s),

Reviewer’s comment No 6

Thank you for the opportunity to review this manuscript. This research seems timely, but having read the manuscript, the study seems like it would benefit from a much more careful consideration of relevant bodies of literature, as well as the articulation of a coherent theoretical/conceptual framework and hypotheses/research questions. Several things could also be done to strengthen the “Materials and Methods” section.

Authors’ response 6: 

We would like to thank the reviewer for their comments and feedback. To strengthen the manuscript, the following changes have been made as advised:

Page 4, line 63 to 73. 

Elevated health related anxiety is understandable during a pandemic. As stated in the literature around health anxiety (Creed and Barsky, 2004; Afifi et al., 2018) the stress due to COVID 19 is situated in the feelings of contracting or having the disease. The triggering of high levels of anxiety and the ensuing coping mechanisms during an event such as the current global pandemic, is aligned with the transactional model of stress and coping as set out by Lazarus and Folkman (1987). This model indicates that behavioural and cognitive coping responses are used by individuals in order to control internal and external stressors. For some individuals these anxiety symptoms may be inherently frightening and/ or misperceived as signs of physical illness, thus increasing overall anxiety levels. When faced with threat social resources can play an important role in an individual’s coping mechanism and ultimately their ability to function (Billings and Moos, 1981; Lazarus and Folkman, 1987).

Page 6, lines 122 - 130

Our overall conceptual model is taken from the core ideas of the transactional model of stress and coping as set out by Lazarus and Folkman (1987). Our conceptual model sets out the empirical relationships among the antecedent, mediating and outcome variables that make up the health anxiety process (Table 1). 

Hypothesis: Age impacts the mediating relationship of COVID 19 health anxiety between locus of control and coping mechanisms. 

Table 1 Model of Health Anxiety Process

Causal antecedent Mediating process Outcome

Locus of control Health anxiety Coping

Internal

Personal responsibility

Social responsibility

Values and commitments

External

Beliefs

Control 

 Worry

Risk

Vulnerability

Preoccupation

Distress

Somatic anxiety Behavioural 

Mindful

Social

Source: Adapted from Lazarus and Folkman (1987)

Reviewer’s comment No 7

One of my chief concerns regarding the manuscript pertain to what I perceive to be a lack of theoretical depth. The study is loosely drawing on a very limited number of sources to make an argument, but there’s no overall, coherent theoretical model or framework guiding the investigation. And, of course, there are no hypotheses or research questions stated.

Authors’ response 7: 

The changes listed previously we hope to address these comments and strengthen the theoretical depth of the manuscript.

Reviewer’s comment No 8: On a related note, the authors are testing a model via structural equation modelling procedures but have not presented and justified an initial conceptual model.

Authors’ response 8: 

The following changes have been made: Page 6, lines 122 – 130; Our overall conceptual model is taken from the core ideas of the transactional model of stress and coping as set out by Lazarus and Folkman (1987). Our conceptual model sets out the empirical relationships among the antecedent, mediating and outcome variables that make up the health anxiety process (Table 1). 

Hypothesis: Age impacts the mediating relationship of COVID 19 health anxiety between locus of control and coping mechanisms. Table 1 Model of Health Anxiety Process

Causal antecedent Mediating process Outcome

Locus of control Health anxiety Coping

Internal

Personal responsibility

Social responsibility

Values and commitments

External

Beliefs

Control Worry

Risk

Vulnerability

Preoccupation

Distress

Somatic anxiety Behavioural 

Mindful

Social

Source: Adapted from Lazarus and Folkman (1987)

Reviewer’s comment No 9

The authors mentioned recruitment via social media, but how was the study presented to prospective participants? 

Authors’ response 9: 

Thank you for drawing this to our attention. On pages 7-8 lines 133-142 now read as follows: ‘The Centre for Excellence in Research at the Goa Institute of Management, India was consulted regarding the research protocol. As the survey was anonymous, institutional ethics approval was not required. The Qualtrics link was shared through social media (WhatsApp and Facebook), participants were advised of the purpose and nature of the study. No incentives were offered for participation. Particular reference was made to the importance of studying the impact of COVID -19 on their levels of health anxiety, well-being and coping mechanisms. Participants were informed that the survey would take approximately 10 minutes to complete. Participation in the study was voluntary and anonymous with no personal identifiable data captured. Participants were not permitted to skip any of the items; however, they had a choice to abandon their response at any point.’

Reviewer’s comment No 10

Were there incentives offered? 

Authors’ response 10: 

No incentives were offered. On pages 7 and 8 of manuscript lines 136-137 now state ‘No incentives were offered for participation.

Reviewer’s comment No 11

 Moreover, how do the demographics of this convenience sample compare to official demographics for the population they come from? 

Authors’ response 11: 

The majority of survey participants were in the age group of 18-45 which showed a similar trend to census data as found using the links below: 

https://censusindia.gov.in/Census_Data_2001/India_at_glance/broad.aspx

http://mospi.nic.in/sites/default/files/reports_and_publication/statistical_publication/social_statistics/WM17Chapter1.pdf

India Demographics Profile (indexmundi.com) 

Reviewer’s comment No 12

Why is the nature of the sample not discussed as a limitation later in the paper?

Authors’ response 12: 

We would like to thank the reviewer for bringing this to our attention. The limitations section on pages 16 and 17 (lines 303-309) now read as: ‘. The findings set out in this paper are preliminary, and further work is required to fully understand the differential impact of these different coping strategies on health and wellbeing and the role of uncertainty. In addition, the small sample size limits the generalisability of the findings. The participant recruitment using social media and the requirement to complete survey in English language are further limitations the study. Furthermore, a number of scales were created specifically for this study and have not been fully validated, though the internal consistency of the measures is encouraging.’ 

Reviewer’s comment No 13

Additionally, for two of the three measures adapted from other studies, there are no citations.

Authors’ response 13: 

The missing citations have now been included as below:

Coping mechanisms (adapted from ways of coping check list (WCCL) (Folkman et al., 1986))

Locus of control (adapted from the Rotter internal-external locus of control scale (Rotter, 1966)

Reviewer’s comment No 14

Further, for the measure that the authors constructed themselves – the one on “Coping” – there is no analysis of validity reported. 

There is a Cronbach alpha score reported, which seems quite low (the author indicate in “Limitations” that the internal consistency results are encouraging, but the findings do not quite support this conclusion). 

There’s no explanation from the authors for the low numbers. 

Moreover, where did these dimensions of coping that the authors have identified come from? 

Authors’ response 14

We would like to thank the reviewer for drawing this to our attention. The following changes have been made to the manuscript.

Page 11, lines 183-199156-175 now read as: ‘Exploratory factor analyses were undertaken on the 10 coping items in order to test for the smallest number of interpretable latent factors. An initial estimation yielded 3 factors with eigenvalues exceeding unity, accounting for 52.6% of the total variance. The internal consistency and the items within each factor are listed below:

 Cope 1: Behavioural coping (α =0.623)

• I am thinking of learning something new 

• I feel happy for having more time to be with my family 

• I spend time reading 

• I am taking a good rest 

Cope 2: Mindful Coping (α =0.522)

• I’m practicing mediation to help me cope 

• I have started exercises and yoga at home 

• I am having an ability to resist thoughts of illness 

Cope 3: Coping through social media (α =0.511)

• My social medium usage has gone up 

• I’m using technology to connect with my loved ones 

• I read and enjoy humorous messages and share with others’

Page 11, lines 200-201 now read as: ‘As the Cronbach alpha internal consistency measures are poor to questionable further analysis was performed to determine the reliability (George & Mallery, 2003).’

Pages 11-12, lines 206-207 now read as ‘Information RMSEA, S_RMR, CD, TLI and CFI of the individual measures are provided in Table 3, demonstrating that the predicted model is an acceptable fit (Byrne, 2016; Kline, 2016).’

Page 12 line 210 now starts with ‘Reliability is also illustrated by the items which…’

Reviewer’s comment No 15

Why is there no reference to the coping-related literature in psychology, public health, communication/media, etc. (with the exception of one to Folkman et al.’s work in the Discussion section)? 

The authors seem to talk at some length about social media use in the conclusion but have made no reference to the literature on social media use.

Authors’ response 15:

We would like to thank the reviewer for the opportunity to add relevant literature to the manuscript as follows: 

Pages 4-5. lines 63-98 now read as ‘Elevated health related anxiety is understandable during a pandemic. As stated in the literature around health anxiety (Afifi et al., 2018; Creed and Barsky, 2004; Jungmann and Witthoft, 2020) the stress due to COVID 19 is situated in the feelings of contracting or having the disease. The triggering of high levels of anxiety and the ensuing coping mechanisms during an event such as the current global pandemic, is aligned with the transactional model of stress and coping as set out by Lazarus and Folkman (Lazarus and Folkman, 1987). This model indicates that behavioural and cognitive coping responses are used by individuals in order to control internal and external stressors. For some individuals these anxiety symptoms may be inherently frightening and/ or misperceived as signs of physical illness, thus increasing overall anxiety levels. When faced with threat social resources can play an important role in an individual’s coping mechanism and ultimately their ability to function (Billings and Moos, 1981; Lazarus and Folkman, 1987). Of course, the pandemic is an inherently uncertain situation and research indicates that difficulties coping with uncertainty (known as intolerance of uncertainty - Dugas et al., 1997) are an important transdiagnostic mechanism in a range of anxiety disorders, including health anxiety (Fergus and Valentiner, 2011). Writing about the 2009 H1N1 pandemic Taha and colleagues report that greater intolerance of uncertainty was related to lower appraisals of self‐ and other control, which in turn was associated with low rates of problem‐focused coping and greater reports of pandemic related anxiety (Taha et al., 2014). Furthermore, they report that people with high levels of intolerance of uncertainty were more likely to perceive the pandemic as threatening and were more likely to use emotion‐focused coping strategies, and both of which predicted elevated levels of anxiety. Similarly, in a recent study by Rettie and Daniels, conducted in the context of the COVID 19 pandemic, it was demonstrated using mediation modelling that maladaptive coping responses partially mediated the relationship between intolerance of uncertainty and both anxiety and depression scores (Rettie and Daniels, 2020).

In an effort to reduce the anxiety and gain certainty about health status, individuals will often engage in safety-seeking behaviours, including searching for health related information on the internet (Abramowitz et al., 2002). Whilst of course the use of safety behaviours in the presence of actual threat, such as a pandemic, is essential to maintain survival, excessive and inflexible use of such behaviours has been observed to maintain anxiety disorder symptoms (Salkovskis, 1991). Safety behaviours associated with health anxiety include seeking reassurance from external sources, including doctors, social media and the internet searches (Abramowitz et al., 2002). These behaviours may be employed by an individual to reduce the perception of threat which in turn may create a short-term reduction in health anxiety (Abramowitz & Moore, 2007). However such safety behaviours are negatively reinforcing and maintain anxiety symptoms in the long-term (Salkovskis et al., 2003) and maintain and exacerbate beliefs that health-related uncertainty cannot be tolerated (Olatunji et al., 2011), creating a self-perpetuating cycle of health anxiety symptoms (Abramowitz et al., 2007).’

Reviewer’s comment No 16

Plus, there is no justification for the inclusion of all the variables mentioned in Table 5, reporting findings from a linear regression predicting health anxiety.

Authors’ response No 16

A recent systematic review by Xiong et al (2020) reported that risk factors associated with psychological distress included female gender, younger age group (≤40 years), presence of chronic/psychiatric illnesses, unemployment, student status, and frequent exposure to social media/news concerning COVID-19. This combined with the theoretical framework informed the choice of variables.

Reviewer’s comment No 17

Additionally, the authors make two references to the Bhopal disaster, which occurred back in 1984, but never provide any additional information about it; so international audiences may not be able to appreciate these references and the significance of that tragic event.

Authors response No 17: 

We would like to thank the reviewer for allowing us to provide more contextual information, In the Introduction section on pages 5-6 lines 102-116 now read as ‘This is even more relevant in a developing country such as India as seen during the Bhopal gas tragedy. In December 1984, following a leakage of a highly hazardous chemical, the local population reported symptoms of suffocation, intense irritation, and vomiting. Murthy highlights the significance of this disaster thus (Murthy, 1990):

“The Bhopal disaster is of importance in the relevant literature for a number of reasons. First, it is one of the largest man-made disasters in a developing country. Second, the disaster effects were a combination of both the chemical substances inhaled and the psychological effects. Third, no formal mental health infrastructure was available to provide post-disaster mental health care, and this led to the development of the innovative approaches to care. Fourth, this disaster has been the subject of intense study, both cross-sectionally and longitudinally, from physical and mental health viewpoints.”

The tragedy highlights the importance of both identifying and then treating mental health sequalae of emergencies in developing countries. This is also, highlighted in the WHO Mental Health Gap Action Programme (mhGAP) which aims to scale up services in low- and middle-income countries as there is a lack of access to appropriate treatments (World Health Organisation, 2010).’

Additional text in conclusions section on pages 17-18 lines 324-333 now read as ‘As Murthy says, though disasters are a challenge in every country, for the affected populations as well as the mental health professionals, they represent special challenges and opportunities in developing countries (Murthy, 2014). The Bhopal disaster was the first disaster in India to be studied systematically for the mental health effects with three decades of mental health initiatives. The trauma of the survivors of the Bhopal gas disaster has continued. In the absence of systemic intervention, the impact on the mental health of the affected residents has been reported to have worsened involving both neurotoxicity and a range of mental disorders (Ranjan Basu and Murthy, 2003). An important lesson learnt is that there is a need for continuous dialog with the survivors, to both understand their perceptions and needs, as well as for wide use of “self-care” measures for mental health. The findings of this study echo the significance of this lesson.’

Reviewer’s comment No 18

On p. 9 “In the current pandemic, data from China indicate severe mental health symptoms in the affected regions (Dong and Bouey, 2020), fuelled by vast amount of information available via social and other media sources (Mertens et al., 2020)”: 

(i)symptoms such as? 

(ii) What do these studies find? These seem like very general statements.

Authors’ response No 18: 

We would like to thank the reviewer for bringing this to our attention. We have added more specificity to the text (page 3, lines 42-53) whilst also updating the references as below:

‘In a recent systematic review (Xiong et al., 2020) high rates of mental health symptoms were reported in studies assessing the general population in China, Spain, Italy, Iran, USA, Turkey, Nepal, and Denmark during the first 6 months of the COVID-19 pandemic. These mental health symptoms included anxiety (6.33% to 50.9%), depression (14.6% to 48.3%), post- traumatic stress disorder (7% to 53.8%), non-specific psychological distress (34.43% to 38%), and stress (8.1% to 81.9%). Recent studies have also reported that increased exposure to social media and/or news information concerning the pandemic was positively associated with increased symptoms of anxiety (Gao et al., 2020; Mertens et al., 2020; Moghanibashi-Mansourieh, 2020). These data are in keeping with earlier reports of increased stress and vulnerability at the population level in response to previous outbreaks, particularly in those who are less resilient (Balinska and Rizzo, 2009; Blendon et al., 2004; Hong and Collins, 2006; Leung et al., 2004; Rosling and Rosling, 2003; Taha et al., 2014) ).’

Reviewer’s comment No 19

“Heightened exposure to pandemic related information may influence risk perception and play a role in shaping coping mechanisms (Gilman, 2010)”: Play a role how? And what is it about “heightened exposure” that matters here?

Authors’ response No 19: 

This sentence has been deleted now to improve readability and flow of the paper. 

Reviewer’s comment No 20

On p. 10, “Unfortunately these behaviours can in turn negatively reinforce health anxiety and beliefs that health-related uncertainty cannot be tolerated (Olatunji et al., 2011), creating a self-perpetuating cycle of health anxiety symptoms (Abramowitz et al., 2007b)”: This is an over-generalization. This could happen, but obviously not for everyone. In many cases health-related information seeking can help reduce uncertainty, too, no? I wouldn’t base my argument on just one study.

Authors’ response No 20

Pages 4-5. lines 63-98 now read as ‘Elevated health related anxiety is understandable during a pandemic. As stated in the literature around health anxiety (Afifi et al., 2018; Creed and Barsky, 2004; Jungmann and Witthoft, 2020) the stress due to COVID 19 is situated in the feelings of contracting or having the disease. The triggering of high levels of anxiety and the ensuing coping mechanisms during an event such as the current global pandemic, is aligned with the transactional model of stress and coping as set out by Lazarus and Folkman (Lazarus and Folkman, 1987). This model indicates that behavioural and cognitive coping responses are used by individuals in order to control internal and external stressors. For some individuals these anxiety symptoms may be inherently frightening and/ or misperceived as signs of physical illness, thus increasing overall anxiety levels. When faced with threat social resources can play an important role in an individual’s coping mechanism and ultimately their ability to function (Billings and Moos, 1981; Lazarus and Folkman, 1987). Of course, the pandemic is an inherently uncertain situation and research indicates that difficulties coping with uncertainty (known as intolerance of uncertainty - Dugas et al., 1997) are an important transdiagnostic mechanism in a range of anxiety disorders, including health anxiety (Fergus and Valentiner, 2011). Writing about the 2009 H1N1 pandemic Taha and colleagues report that greater intolerance of uncertainty was related to lower appraisals of self‐ and other control, which in turn was associated with low rates of problem‐focused coping and greater reports of pandemic related anxiety (Taha et al., 2014). Furthermore, they report that people with high levels of intolerance of uncertainty were more likely to perceive the pandemic as threatening and were more likely to use emotion‐focused coping strategies, and both of which predicted elevated levels of anxiety. Similarly, in a recent study by Rettie and Daniels, conducted in the context of the COVID 19 pandemic, it was demonstrated using mediation modelling that maladaptive coping responses partially mediated the relationship between intolerance of uncertainty and both anxiety and depression scores (Rettie and Daniels, 2020).

In an effort to reduce the anxiety and gain certainty about health status, individuals will often engage in safety-seeking behaviours, including searching for health related information on the internet (Abramowitz et al., 2002). Whilst of course the use of safety behaviours in the presence of actual threat, such as a pandemic, is essential to maintain survival, excessive and inflexible use of such behaviours has been observed to maintain anxiety disorder symptoms (Salkovskis, 1991). Safety behaviours associated with health anxiety include seeking reassurance from external sources, including doctors, social media and the internet searches (Abramowitz et al., 2002). These behaviours may be employed by an individual to reduce the perception of threat which in turn may create a short-term reduction in health anxiety (Abramowitz & Moore, 2007). However such safety behaviours are negatively reinforcing and maintain anxiety symptoms in the long-term (Salkovskis et al., 2003) and maintain and exacerbate beliefs that health-related uncertainty cannot be tolerated (Olatunji et al., 2011), creating a self-perpetuating cycle of health anxiety symptoms (Abramowitz et al., 2007).’

Reviewer’s comment No 21

On p. 10, “This is even more relevant in a developing country such as India as seen during the Bhopal gas tragedy in 1984 (Murthy, 2014)”: Why more relevant in a developing country?

Authors’ response No 21: 

Pages 5-6, lines 103-116 now read as ‘In December 1984, following a leakage of a highly hazardous chemical, the local population reported symptoms of suffocation, intense irritation, and vomiting. Murthy highlights the significance of this disaster thus (Murthy, 1990):

“The Bhopal disaster is of importance in the relevant literature for a number of reasons. First, it is one of the largest man-made disasters in a developing country. Second, the disaster effects were a combination of both the chemical substances inhaled and the psychological effects. Third, no formal mental health infrastructure was available to provide post-disaster mental health care, and this led to the development of the innovative approaches to care. Fourth, this disaster has been the subject of intense study, both cross-sectionally and longitudinally, from physical and mental health viewpoints.”

The tragedy highlights the importance of both identifying and then treating mental health sequalae of emergencies in developing countries. This is also, highlighted in the WHO Mental Health Gap Action Programme (mhGAP) which aims to scale up services in low- and middle-income countries as there is a lack of access to appropriate treatments (World Health Organisation, 2010).’

Reviewer’s comment No 22

On p. 11, “For this anonymised survey institutional approvals were sought as appropriate”: Sought, but were they actually obtained? If the survey was anonymous, this study would likely get “exempted” status, so I am just suggesting clarification here.

Authors’ response No 22: this has been previously addressed in authors’ response No 2

Reviewer’s comment No 23

On p. 11, was the measure for “Coping Mechanisms” which was created by the authors examined for its validity? If so, how? What dimensions of coping are captured? There’s a significant literature on coping that has not been consulted/used here.

Authors’ response No 23: 

Thank you for drawing this to our attention. The section on how measures were adapted on page 8 (lines 149-156) now read as ‘In order to ensure all measures were appropriate to the sociocultural setting in which the study was conducted, the following steps were followed: Health Anxiety scale, 4 behavioural experts from different institutes including Goa Institute of Management were asked to suggest appropriate items from the Short Health Anxiety Inventory (Abramowitz et.al 2007a) to capture health anxiety. Based on the responses, 5 items were re-phrased from the inventory and 1 item (I have started to feel worrisome about my own loved one’s health) was added as a new item. For the Coping Mechanisms and Locus of Control items the same experts were asked to select the items in a similar way. Repetitive items were excluded and some of the items were re-phrased for easy reading.’ 

Reviewer’s comment No 24

Also, what is the source for the “Locus of Control” measure? Please provide citations.

Authors’ response No 24: 

Locus of control (adapted from the Rotter internal-external locus of control scale (Rotter, 1966). Citations have been updated in the text. 

Reviewer’s comment No 25

On p. 16, how did these age groups get defined (e.g., 18-31, 32-45, 46+)? Based on what?

Authors’ response No 25

Thank you for this query. Pages 13-14, lines 227-238 now read a ‘Regarding sample size a desirable goal for an excellent model is to have a ratio of 20:1 for the number of respondents to the number of parameters in the model. A good model should have a ratio of 10:1 and if the ratio is less than 5:1 the model is said to be poor (Humble, 2020; Kline, 1998). With 6 parameters in the model (internal LOC, external LOC, Anxiety, Cope 1, Cope 2, Cope 3) then the ratios would be excellent for 120 respondents and good for 60 respondents. The categories in this model have the following number of respondents illustrating a good sample size for each of the models:

• 18 to 31 years -101 respondents

• 32 to 45 years – 75 respondents 

• over 46 years – 58 respondents 

Using these three age categories, 3 standardized coefficient structural path models using the latent factor scores from the proceeding analysis were constructed (Figure 2, 3 and 4).’

Reviewer #2: 

Reviewer’s comment No 26

First impressions of the study:

The research appears to be taking initial steps to identify mental health factors related to the COVID-19 pandemic and initial stages of lockdown conditions in India. There is limited data on this phenomenon globally and would add to the growing database – especially as preliminary findings from South Asia. The study has potential to add to international research, and the contribution of knowledge to facilitate comparisons between South Asia and rest of the world with reference to coping mechanisms and health anxiety. The manuscript appears to be relevant and interesting.

Authors’ response No 26: 

We would like to thank the reviewer for the positive feedback which reaffirms why we wanted to publish this work to add to the scientific evidence.

Reviewer’s comment No 27

Title, Abstract, References:

Full title is informative, relevant, to the point and conveys key features of the article. The title is likely to spark interest in the scientific mental health community – given the present context of living with COVID 19 conditions and the need to inform provisions for services in mental health in South Asian and Globally.

The objectives of the study are clearly stated within the abstract, with methods outlined, and results and conclusions section which is clearly aligned with aim. The methods section within abstract can be improved by indicating the measures used within the study.

References cited and listed correctly. Recommended further inclusion of recent international research articles on mental health related to coping and health anxiety within discussion during COVID -19 as points of comparison or discussion would improve impact of article. Recent literature has been published related to COVID-19 and health anxiety experienced globally, for e.g. Germany (Jungmann & Witthoft, 2020), UK (Rettie & Daniels, 2020), Turkey (Ozdin & Ozdin, 2020), US (Tull, et al., 2020), Canada (Taylor, Landry, Paluszek, Rachor, & Asmundson, 2020).

Authors’ response No 27: 

We would like to thank the reviewer for their detailed feedback and also for providing up to date references, some of which have been added.

Reviewer’s comment No 28

The research topic has been introduced by placing the study well within the research context with adequate literature. The importance of the current research has been established and the purpose of the study has been clearly stated. However, the first study aim appears to be general and needs to be specified and revised to state “the study aims to explore the psychological impact of COVID-19 on health anxiety”.

Logical progression in flow of thought observed with enough information to understand the research. Specialized terminology such as locus of control has been explained, and statement of population has been provided. Structure and quality of writing is satisfactory.

Authors response No 28: 

Lines 120-122 now read as ‘This study aims to explore the psychological impact of COVID 19 on health anxiety and the coping strategies used by the adult population in India during the first week of lockdown as a result of COVID 19 pandemic.’

Reviewer’s comment No 29

The process of study subject selection is clearly provided; however, it was observed that ethical clearance to conduct study was not obtained by the authors. Manuscript states that institutional clearance was obtained – although not specifically stated from where. Please provide this information for clarity of ethics approval for study conducted in the field of psychology/psychiatry among human participants.

Authors’ response No 29: 

This has been addressed in authors’ response no 2.

Reviewer’s comment No 30

The procedure does not indicate the language in which the study was conducted – this needs to be indicated for reproducibility of the study and also to correctly gauge study population and thereby, any implications. 

Authors’ response No 30: 

The study was carried out in the English language. This has been added to the following sections of the manuscript:

Methods Page 8, lines 144-145 now read as ' Data were collected in the English language during week 1 (24th March – 30th March 2020) of lockdown in India.’

Limitations: ‘The participant selection using social media and the requirement to complete survey in English language is a limitation to the study.’

In addition, Rotter’s scale has previously been used in India in English. https://ijip.in/wp-content/uploads/ArticlesPDF/article_fc5ca0ec15dcf1348d987769a0d9cc06.pdf

And whilst some of the measures e.g. Rotter’s scale have a Hindi version (Kohli, S., Batra, P., & Aggarwal, H. K. (2011). Anxiety, locus of control, and coping strategies among end-stage renal disease patients undergoing maintenance hemodialysis. Indian journal of nephrology, 21(3), 177.), they are not available in all 122 major languages and 1599 other languages in India (Census of India of 2001)

Reviewer’s comment No 31

Further, the measures listed in study gives rise to the requirement of following clarifications:

Is the provided reference correct for the ‘Health anxiety scale’? – The citation given refers to the Psychometric properties of the Short Health Anxiety Inventory, and not the six-item scale used in this study.

Authors’ response No 31

As indicated in Table 2, the Health anxiety scale used in this study was an adapted version of the Short Health Anxiety Inventory. The citation (reference of Abramowitz et a 2007 is for the Short Health Anxiety Inventory. 

Reviewer’s comment No 32

Was the health inventory adapted/validated to the Indian context? If so, the correct reference should be provided. Was there a method used to generate the questionnaire items for the scales?

Authors’ response No 32

Thank you for drawing this to our attention. The section on how measures were adapted on page 8 (lines 149-156) now read as ‘In order to ensure all measures were appropriate to the sociocultural setting in which the study was conducted, the following steps were followed: Health Anxiety scale, 4 behavioural experts from different institutes including Goa Institute of Management were asked to suggest appropriate items from the Short Health Anxiety Inventory (Abramowitz et.al 2007a) to capture health anxiety. Based on the responses, 5 items were re-phrased from the inventory and 1 item (I have started to feel worrisome about my own loved one’s health) was added as a new item. For the Coping Mechanisms and Locus of Control items the same experts were asked to select the items in a similar way. Repetitive items were excluded and some of the items were re-phrased for easy reading.’ 

Reviewer’s comment No 33

How was the sample size estimated for this study?

Authors’ response No 33

Thank you for this query. Pages 13-14, lines 227-238 now read a’ Regarding sample size a desirable goal for an excellent model is to have a ratio of 20:1 for the number of respondents to the number of parameters in the model. A good model should have a ratio of 10:1 and if the ratio is less than 5:1 the model is said to be poor (Humble, 2020; Kline, 1998). With 6 parameters in the model (internal LOC, external LOC, Anxiety, Cope 1, Cope 2, Cope 3) then the ratios would be excellent for 120 respondents and good for 60 respondents. The categories in this model have the following number of respondents illustrating a good sample size for each of the models:

• 18 to 31 years -101 respondents

• 32 to 45 years – 75 respondents 

• over 46 years – 58 respondents 

Using these three age categories, 3 standardized coefficient structural path models using the latent factor scores from the proceeding analysis were constructed (Figure 2, 3 and 4).’

Reviewer’s comment No 34

Minor errors- Typos observed in item 5 ‘mediation’ and item 8 ‘social medium’ under the coping mechanisms scale.

Authors’ response No 34

Thank you for highlighting the same. These have now been corrected to read as ‘meditation’ and ‘social media’ in the manuscript in Table 2 under ‘Coping Mechanisms’ on page 9.

Reviewer’s comment No 35

In general, data has been presented well, with the text provided adding to the tabulated/illustrated results. If there is a recognized cut off mark for health anxiety it is suggested to indicate the percentage and levels of health anxiety observed in the study sample within descriptive data.

Authors’ response No 35: 

There is currently no recognised cut off mark for health anxiety scale.

Reviewer’s comment No 36

In process of answering the study questions, the authors have also taken efforts towards the establishment of construct validity of the author generated question items/factors. However, correct interpretation in reporting encouraged. For instance, internal consistency reliability for the author generated question items in coping mechanisms and locus of control are less than 7, which is questionable. Further, the three factors, within the coping scale (cope 1,2 and 3) also report alpha values less than 7 and 6, which are usually interpreted as ‘poor to questionable’ (George & Mallery, 2003).

Line 103, the authors provide values to what figured constitute a good fit. However, model fit has been declared a ‘good’ fit, whilst the findings are not reporting the same. The model fit are adequate or borderline RMSEA > 0.06, S.RMR = 0.008 (not less), CFI <0.9. Consider revising to state, ‘acceptable fit’ should the literature support the findings (Byrne, 2016; Kline, 2016).

Authors’ response No 36

We would like to thank the reviewer for these extremely helpful comments and feedback. The following changes have been made to the manuscript:

Page 11, lines 200-201 now read as: ‘As the Cronbach alpha internal consistency measures are poor to questionable further analysis was performed to determine the reliability (George & Mallery, 2003).’

Pages 11-12, lines 206-207 now read as ‘Information RMSEA, S_RMR, CD, TLI and CFI of the individual measures are provided in Table 3, demonstrating that the predicted model is an acceptable fit (Byrne, 2016; Kline, 2016).’

Page 12 line 210 now starts with ‘Reliability is also illustrated by the items which…’

Reviewer’s comment No 37

Within the linear regression data, it was observed gender was not included as an independent variable. Consider including as the relationship with gender and health anxiety would also add to the available database on the subject.

Authors’ response No 37

We would like to thank the reviewer for this suggestion. The linear regression has been carried out again to include gender and the new table is Table 6 thus and was not found to be statistically significant:

Coefficientsa

Model Unstandardized Coefficients Standardized Coefficients t Sig.

 B Std. Error Beta 

 (Constant) .354 .224 1.580 .115

 Age of the participants -.312 .083 -.233 -3.752 .000

 Gender (female = 1) .019 .135 .009 .141 .888

 Participant lives with their family -.154 .211 -.044 -.730 .466

 Participant from a metro or non-metro city .105 .133 .048 .788 .431

 Internal Locus of Control .120 .061 .129 1.970 .050

 External Locus of Control .187 .056 .210 3.321 .001

 Cope 1: Behavioural Coping .101 .060 .114 1.691 .092

 Cope 2: Mindful Coping .083 .060 .090 1.379 .169

 Cope 3: Coping through Social Media .173 .057 .192 3.020 .003

a. Dependent Variable: Exploratory health anxiety latent variable. P[F(9, 224) = 6.328)< 0.001], R2 = 0.203

Reviewer’s comment No 38

Meaningful results have been discussed well and placed within context without over interpretation. Conclusions answer the aims of the study (aim one in introduction section needs to be specified). Limitations of study have been clearly indicated. Following minor modifications are suggested,

Line 211 - … “the use of other ‘adaptive’ behavioral and mindful techniques..”]

Authors’ response No 38

We acknowledge the feedback. The phrase ‘behavioural and mindful coping strategies’ referred to the groups operationalised within ‘coping strategies’ defined earlier in the paper and therefore has not been changed.

Reviewer’s comment No 39

235-240 – Revise and summarize the final paragraph to increase impact.

Line 200 – addition of other references that are also confirming that young adults such as (Gerolimatos & Edelstein, 2012) report increased levels of health anxiety would further support the findings.

Authors response No 39

On page 15, lines 279-281 now read as ‘In addition, health-related anxiety was highest in the 18-31 years group which has been previously reported (Gerolimatos and Edelstein, 2012).’

Page 17 lines 311-323 now read as ‘The data used for the analysis give a unique insight into levels of health anxiety and patterns of coping strategies in an adult Indian community sample to the COVID 19 pandemic. The findings also provided important insights into age related differences in response to the pandemic and some preliminary insight into the effectiveness of differential coping mechanisms to mitigate against the anxiety inevitably conferred by this global crisis. It has been shown that younger people were experiencing more health-related anxiety in response to COVID 19 and were more likely to engage in coping through social media than their older counterparts. Tentative support for previous work indicates that mindfulness-based strategies may reduce risk for anxiety potentially by increasing tolerance of the inevitable uncertainty currently experienced. The uniqueness of this global pandemic and the gathering of data in this India context around coping will provide an important point of reference that enhances understanding and knowledge relating to mental health strategies. With a limited number of studies around COVID 19 emanating from India with a focus on mental health makes this an internationally important study.’

Reviewer’s comment No 40

Study design poses limitations however, acceptable. Rigorous statistical analysis conducted to establish construct validity of measure, and results for the study. The study confirmed coping patterns and health anxiety relationship with age and contributes novel information regarding mental health impact during COVID-19 from South Asia. Article is consistent with itself.

Authors’ response No 40

We appreciate the overall positive feedback including appropriate references which have now been included in the manuscript.

---

## [Decision Letter · Decision Letter 1]

12 Mar 2021

PONE-D-20-23653R1

Health anxiety, coping mechanisms and COVID 19: an Indian community sample at week 1 of lockdown

PLOS ONE

Dear Dr. Sharma,

Thank you for submitting your manuscript to PLOS ONE. After careful consideration, we feel that it has merit but does not fully meet PLOS ONE’s publication criteria as it currently stands. Therefore, we invite you to submit a revised version of the manuscript that addresses the points raised during the review process.

We look forward to receiving your revised manuscript.

Kind regards,

Johnson Chun-Sing Cheung, D.S.W.

Academic Editor

PLOS ONE

Journal Requirements:

Reviewers' comments:

Reviewer #1: 

I’d like to commend the authors for the significant amount of work they clearly put into this revision. They have addressed many of the comments, questions, and issues that the reviewers previously raised.

Although I continue to think this manuscript could make for a useful contribution and I recognize the effort put into more clearly articulating a theoretical model underlying the analysis (an issue I raised in my review of the original document), I do have a small number of questions and comments that I’d like them to consider:

1-The authors have reported that the fit of their original conceptual model to the data is satisfactory. That’s great and the scores look encouraging, but I wonder if they considered making any revisions to their original model, based on their initial SEM analyses and modification indices. If not, why not? If so, but they ultimately decided to stick with their original model, why was that the case? If they did make changes, what were those?

2-It would be helpful to include tables containing the regression weights of the hypothesized theoretical model, for both direct and indirect effect scores. A discussion of possible indirect effects would be helpful, too.

3-Three different age groups are compared. However, it seems to me that a proper multigroup SEM analysis is required to ascertain if the hypothesized structure among the key variables in the theoretical model holds across these groups (or not). Findings in either direction could be useful. At the moment it seems (and please correct me if I am wrong) that there was no formal analysis was done to test differences across age groups.

Reviewer #2: Thank you for the opportunity to re-review this manuscript. All initial reviewer comments have been addressed satisfactorily by the authors.

A final clarification remains regarding Author Response 32-

Has correct methodology been utilized to adapt and validate the study instruments to the Indian cultural context? If so, please cite them within the article. If not, clearly state the failure to use any specific guidelines within the limitations of the study. At present, the limitations note that several scales were adapted, however, not “fully” validated—a potential misrepresentation as adherence to any recognized method or guidelines for cross-cultural adaptation and validation is not observed within study.

---

## [Author Response · Author response to Decision Letter 1]

26 Mar 2021

To 

The Academic Editor and Reviewers 

We would like to thank you for your comments. These are addressed in an itemised manner below: 

Academic Editor’s comment

Comment No 1: Journal Requirements:

Please review your reference list to ensure that it is complete and correct. If you have cited papers that have been retracted, please include the rationale for doing so in the manuscript text or remove these references and replace them with relevant current references. Any changes to the reference list should be mentioned in the rebuttal letter that accompanies your revised manuscript. If you need to cite a retracted article, indicate the article’s retracted status in the References list and also include a citation and full reference for the retraction notice.

Authors’ response No 1: We would like to thank the academic editor for drawing to our attention the reference section. The appropriate formatting has now been used and references checked as advised.

Reviewers' comments:

Reviewer #1:

 Comment No 2: I’d like to commend the authors for the significant amount of work they clearly put into this revision. They have addressed many of the comments, questions, and issues that the reviewers previously raised.

Although I continue to think this manuscript could make for a useful contribution and I recognize the effort put into more clearly articulating a theoretical model underlying the analysis (an issue I raised in my review of the original document), I do have a small number of questions and comments that I’d like them to consider:

Authors’ response No 2: We would like to thank the reviewer for their appreciative comments

Comment No 3: The authors have reported that the fit of their original conceptual model to the data is satisfactory. That’s great and the scores look encouraging, but I wonder if they considered making any revisions to their original model, based on their initial SEM analyses and modification indices. If not, why not? If so, but they ultimately decided to stick with their original model, why was that the case? If they did make changes, what were those?

Authors’ Response No 3: We would like to thank the reviewer for the opportunity to provide additional information about the models as informed by the SEM analyses. The manuscript on page 11 lines 193-200 now read as: Information RMSEA, S_RMR, CD, TLI and CFI of the individual measures are provided in Table 3 (Model I), demonstrating that the predicted model is an acceptable fit (Byrne, 2016; Kline, 2016).

Table 3 Fit Indices of the whole model 

 Fit Index 

 χ2 df RMSEA S-RMR CD TLI CFI

Model I 429.934 225 0.063 0.08 0.920 0.812 0.833

Model II 194.749 194 0.004 0.055 0.926 0.999 0.999

It was considered whether any revisions to this model should be made given the acceptable fit indices. Modification indices recommended the inclusion of some disturbance covariance terms between item variables. This resulted in a better data to model fit (Model II) and confirmed our confidence in the structure. 

Comment No 4: It would be helpful to include tables containing the regression weights of the hypothesized theoretical model, for both direct and indirect effect scores. A discussion of possible indirect effects would be helpful, too.

Authors’ response No 4: The following details have been added to the manuscript on page 12 lines 207-211: ‘There are significant indirect effects mediated by health anxiety from internal and external locus of control to all of the coping strategies as illustrated in the table above. This indicates that a person’s internal and external locus of control have an effect on how they cope. 

Table 4 Regression standardized structural coefficients

 Endogenous variable

Exogenous variable 

 Health Anxiety Behavioural Coping Mindful Coping Coping through Social Media

Internal LoC 

Direct effect 0.304*** 

Indirect effect 0.102* 0.067* 0.146**

External LoC 

Direct effect 0.247*** 

Indirect effect 0.083* 0.054* 0.119*

Health Anxiety 

Direct effect 0.336*** 0.220** 0.480***

p***<0.001, p**<0.01, p*<0.05. Measurement path coefficients all sig p<0.05

Comment No 5: Three different age groups are compared. However, it seems to me that a proper multigroup SEM analysis is required to ascertain if the hypothesized structure among the key variables in the theoretical model holds across these groups (or not). Findings in either direction could be useful. At the moment it seems (and please correct me if I am wrong) that there was no formal analysis was done to test differences across age groups.

Authors’ response No 5: We have added the following details on pages 13-14 lines 232-239: ‘A test for measurement invariance was undertaken on age subgroups comparing the unconstrained model with all parameters free across groups, to one with coefficients invariant. The chi-square and fit indices difference (��2(14) = 10.942, p=0.691; �RMSEA=0.036; �CFI=0.027) demonstrates the models measurement invariance (Rutkowski, & Svetina, 2014; Chen, 2007). The equality of the model was also assessed through metric invariance tests. These tests of the structural model gave non-significant results across all structural path parameters indicating that the constructs within this model hold across groups (Cheung & Rensvold, 2002; Gregorich, 2006).’

Reviewer #2: 

Comment No 6: Thank you for the opportunity to re-review this manuscript. All initial reviewer comments have been addressed satisfactorily by the authors.

A final clarification remains regarding Author Response 32-

Has correct methodology been utilized to adapt and validate the study instruments to the Indian cultural context? If so, please cite them within the article. If not, clearly state the failure to use any specific guidelines within the limitations of the study. At present, the limitations note that several scales were adapted, however, not “fully” validated—a potential misrepresentation as adherence to any recognized method or guidelines for cross-cultural adaptation and validation is not observed within study.

Authors’ response No 6: We thank the reviewer for both appreciating the responses and drawing our attention to the above. The limitations section on page 16 lines 303-305 now reads as: ‘A number of scales were adapted for this study. Although the internal consistency for these measures is encouraging, cross cultural adaptation and validation is yet to be undertaken.’

---

## [Editor Report · Decision Letter 2]

6 Apr 2021

Health anxiety, coping mechanisms and COVID 19: an Indian community sample at week 1 of lockdown

PONE-D-20-23653R2

Dear Dr. Sharma,

We’re pleased to inform you that your manuscript has been judged scientifically suitable for publication and will be formally accepted for publication once it meets all outstanding technical requirements.

Kind regards,

Johnson Chun-Sing Cheung, D.S.W.

Academic Editor

PLOS ONE

---

## [Editor Report · Acceptance letter]

8 Apr 2021

PONE-D-20-23653R2 

Health anxiety, coping mechanisms and COVID 19: an Indian community sample at week 1 of lockdown 

Dear Dr. Sharma:

I'm pleased to inform you that your manuscript has been deemed suitable for publication in PLOS ONE. Congratulations! Your manuscript is now with our production department. 

Kind regards, 

on behalf of

Dr. Johnson Chun-Sing Cheung 

Academic Editor

PLOS ONE